# Integrin α2 and β1 Cross-Communication with mTOR/AKT and the CDK-Cyclin Axis in Hepatocellular Carcinoma Cells

**DOI:** 10.3390/cancers14102430

**Published:** 2022-05-14

**Authors:** Mazen A. Juratli, He Zhou, Elsie Oppermann, Wolf O. Bechstein, Andreas Pascher, Felix K.-H. Chun, Eva Juengel, Jochen Rutz, Roman A. Blaheta

**Affiliations:** 1Department of General and Visceral Surgery, Goethe-University, 60590 Frankfurt am Main, Germany; mazen.juratli@ukmuenster.de (M.A.J.); athene99cn@gmail.com (H.Z.); oppermann@em.uni-frankfurt.de (E.O.); wolf.bechstein@kgu.de (W.O.B.); 2Department of General, Visceral and Transplant Surgery, Muenster University Hospital, 48149 Muenster, Germany; birgit.freitag@ukmuenster.de; 3Department of Urology, Goethe-University, 60590 Frankfurt am Main, Germany; felix.chun@kgu.de (F.K.-H.C.); jochen.rutz@kgu.de (J.R.); 4Department of Urology and Pediatric Urology, University Medicine Mainz, 55131 Mainz, Germany; eva.juengel@unimedizin-mainz.de

**Keywords:** hepatocellular carcinoma, AKT-mTOR pathway, integrins, growth, invasion

## Abstract

**Simple Summary:**

Hepatocellular carcinoma (HCC) progression depends on two major processes, tumor growth and invasion. The present study investigated how these events are linked. A panel of HCC cell lines were stimulated with insulin-like growth factor-1 (IGF1) and the biological behavior was evaluated. IGF1 activated the proliferation and invasion cascade by altering the expression level of integrin α subtypes, which were associated with the AKT-mTOR pathway and the CDK-Cyclin axis. We assume that HCC progression is controlled by a fine-tuned network between IGF1 driven integrin signaling, the Akt-mTOR pathway, and the CDK-Cyclin axis. Concerted targeting of these pathways may, therefore, become an innovative option to prevent cancer dissemination.

**Abstract:**

Integrin receptors contribute to hepatocellular carcinoma (HCC) invasion, while AKT-mTOR signaling controls mitosis. The present study was designed to explore the links between integrins and the AKT-mTOR pathway and the CDK-Cyclin axis. HCC cell lines (HepG2, Huh7, Hep3B) were stimulated with soluble collagen or Matrigel to activate integrins, or with insulin-like growth factor 1 (IGF1) to activate AKT-mTOR. HCC growth, proliferation, adhesion, and chemotaxis were evaluated. AKT/mTOR-related proteins, proteins of the CDK-Cyclin axis, focal adhesion kinase (FAK), and integrin-linked kinase (ILK) were determined following IGF1-stimulation or integrin knockdown. Stimulation with collagen or Matrigel increased tumor cell growth and proliferation. This was associated with significant alteration of the integrins α2, αV, and β1. Blockade of these integrins led to cell cycle arrest in G2/M and diminished the number of tumor cell clones. Knocking down the integrins α2 or β1 suppressed ILK, reduced FAK-phosphorylation and diminished AKT/mTOR, as well as the proteins of the CDK-Cyclin axis. Activating the cells with IGF1 enhanced the expression of the integrins α2, αV, β1, activated FAK, and increased tumor cell adhesion and chemotaxis. Blocking the AKT pathway canceled the enhancing effect of IGF on the integrins α2 and β1. These findings reveal that HCC growth, proliferation, and invasion are controlled by a fine-tuned network between α2/β1-FAK signaling, the AKT-mTOR pathway, and the CDK–Cyclin axis. Concerted blockade of the integrin α2/β1 complex along with AKT-mTOR signaling could, therefore, provide an option to prevent progressive dissemination of HCC.

## 1. Introduction

Hepatocellular carcinoma (HCC) is the fifth most common cancer worldwide for men, and the seventh most common cause of cancer death among women. Worldwide, more than 800,000 people are affected by HCC with more than 700,000 deaths each year. The American Cancer Society estimates that 42,810 new cases will be diagnosed with 30,160 deaths in 2020. Since 1980, HCC rates have more than tripled and death rates have more than doubled [1].

Once the tumor has spread to other organs, it is difficult to treat and chemotherapy has been shown to be of limited efficacy. Targeted therapies or the application of immune checkpoint inhibitors have shown promise in improving treatment, but severe side effects and rapid development of resistance hamper their use. Due to the complexity of cellular processes, triggering HCC growth and invasive progression, a successful therapeutic regimen has still not been developed.

The insulin growth factor (IGF) receptor, as well as further tyrosine kinase receptors, play a key role in HCC development and growth. This particularly includes IGF triggered activation of phosphatidylinositol3-kinase (PI3K) and PI3K-related downstream targets with AKT and mechanistic target of rapamycin (mTOR) as prominent growth regulators [2]. Investigations point to a distinct correlation between PI3K/AKT/mTOR pathway activation and malignant clinical outcomes in HCC patients [2]. Indeed, clinical information from the International Cancer Genome Consortium revealed PI3K/AKT/mTOR to be critically involved in the invasion and metastasis of liver cancer [3]. Interaction of AKT with cell cycle regulating proteins were documented as well [4]. This is notable, since up-regulation of cyclin-dependent kinase (CDK) activity correlates with poor HCC prognosis, whereby CDKs may not only serve as prognostic biomarkers [5], but also as promising tumor targets [6].

Subsequent to the establishment of the primary tumor, single cells escape the tumor and start to disseminate. Once released into the blood, tumor cells attach to the vascular wall, transmigrate, interact with basement proteins, and resettle as secondary tumors, restarting the growth program. The process of vascular invasion and HCC metastasis is firmly controlled by adhesion receptors of the integrin family, in particular integrins of the β1 subtype e.g., α1β1 and α5β1 [7]. Recently, the integrins β3 [8] and β4 [9] were demonstrated to not only promote HCC migration but invasion as well.

Although HCC growth and invasion are each controlled by specific intracellular signaling cascades, tumor progression is complex and tightly coordinated. It is, therefore, assumed that AKT/mTOR as well as CDK-Cyclins cross-talk with integrins, such that both tumor growth and metastatic settlement occur in a coordinated manner. Still, the interaction between the integrin pathways with AKT/mTOR and the CDK-Cyclins are not understood in detail. Based on a panel of HCC cell lines, the present investigation dealt with the question of how AKT/mTOR and the CDK-Cyclins modify integrin expression and how integrins contribute to the altered AKT/mTOR and CDK-Cyclin signaling. Whether growth relevant AKT/mTOR is involved in regulating tumor cell adhesion and migration and how adhesion/migration relevant integrins may alter tumor growth and proliferation was also explored.

## 2. Materials and Methods

### 2.1. Cell Lines

HepG2 and Huh7 cell lines were purchased from CLS (Cell Lines Service GmbH, Eppelheim, Germany) and Hep3B from ATCC/LGC (Promochem GmbH, Wesel, Germany). According to the German Central Committee for Biological Safety (ZKBS) the cells can be handled under biosafety level 1. HepG2 was cultured in DMEM/F-12, while Huh7 and Hep3B were cultured in RPMI 1640 (Gibco/Invitrogen, Karlsruhe, Germany). All media were supplemented with 10% fetal bovine serum (FBS), 2% HEPES buffer, 1% GlutaMAX, and 1% penicillin/streptomycin (all obtained from Gibco/Invitrogen). Cells were cultured at 37 °C in a humidified incubator with 5% CO_2_. To activate the cells, soluble IGF1 (100 ng/mL) was used (LONG^®^R3IGF-1; Sigma Aldrich, München, Germany).

### 2.2. Tumor Cell Growth

Cell growth was evaluated using the MTT dye reduction assay, as described recently in [10].

The tumor cells were treated with normal medium (control) or medium enriched with soluble collagen (collagen G; Biochrom, Berlin, Germany; 1, 10, or 100 µg/mL) or with soluble Matrigel (composed of approximately 60% laminin, 30% collagen IV, and 8% entactin; Corning Life Sciences, Tewksbury, MA, USA; 1, 10, or 100 µg/mL). Tumor cell numbers were evaluated after different time periods (24–48 h), using an enzyme-linked immunosorbent assay (ELISA) reader.

### 2.3. Clonogenic Growth Assay

Tumor cells were seeded into 6-well plates at 500 cells per well. They were treated with medium alone (control), or with medium enriched with Matrigel (10 µg/mL) at 37 °C. Following a 1–2 weeks incubation (depending on the cell line used), the plates were fixed with colony staining fixation solution containing glutaraldehyde (6% *v*/*v*) and crystal violet dye (0.5% *w*/*v*) in H_2_O for 30 min at room temperature, and rinsed with tap water. Tumor colonies containing ≥ 50 cells were then counted microscopically.

### 2.4. Integrin Surface Expression

Cancer cells were detached enzymatically (Accutase; PAA Laboratories GmbH, Pasching, Austria), washed with 0.5% BSA (diluted in PBS), and then incubated with the following ready-to-use phycoerythrin (PE) conjugated monoclonal antibodies: Anti-α1 (mouse IgG1; clone SR84); anti-α2 (IgG2a; clone 12F1-H6); anti-α3 (IgG1; clone C3II.1, 20 µL); anti-α4 (mouse IgG1; clone 9F10); anti-α5 (IgG1; clone IIA1); anti-α6 (IgG2a; clone GoH3); anti-β1 (IgG1; clone MAR4); anti-β 3 (IgG1; clone VI-PL2); anti-β4 (IgG2a; clone 439–9B (all obtained from BD Biosciences, San Jose, CA, USA), or anti-αV (IgG1; clone 13C2, Abcam, Cambridge, UK). Integrin expression was measured by a FACSCalibur (BD Biosciences), as described recently in [10]. The surface expression level of the α2, αV, and β1 integrins was additionally evaluated in the presence of soluble collagen (1 or 100 µg/mL) or of soluble Matrigel (10 µg/mL).

### 2.5. Quantitative Polymerase Chain Reaction

To explore integrin α2, αV and β1 gene expression, RNA from HepG2 and Huh7 was isolated using RNeasy Mini Kit (Qiagen, Hilden, Germany), according to [11].

The iTaq Universal SYBR Green Supermix (Bio-Rad Laboratories, Hercules, CA, USA) and gene specific primers (Integrin α2, αV, β1, GAPDH; Bio-Rad Laboratories) were employed: ITGA2 UniqueAssayID: qHsaCID0016134; ITGAV UniqueAssayID: qHsaCID0006233; ITGB1 UniqueAssayID: qHsaCED0005248; GAPDH UniqueAssayID: qHsaCED0038674. RNA levels were normalized with GAPDH and calculated in relative quantification as ∆∆Ct (elative fold change in gene expression).

### 2.6. Western Blot Analysis

Cell cycle regulating protein levels were examined in HepG2 cells. Protein lysates were separated on 7–12% polyacrylamide gel via electrophoresis, and then transferred to PVDF membranes, as described recently in [10]. The following monoclonal antibodies were applied: Anti-CDK1 (IgG1, clone 1); anti-pCDK1/Cdc2 (pY15; IgG1, clone 44/Cdk1/Cdc2); anti-CDK2 (IgG2a, clone 55); anti-Cyclin A (IgG1, clone 25); anti-Cyclin B (IgG1, clone 18); anti-p19 (IgG1, clone 52); Kip1/p27 (IgG1, clone 57) (all obtained from BD Biosciences), and anti-pCDK2 (Thr160; Cell Signaling, Danvers, MA, USA). Antibodies against the mTOR pathway were as follows: Anti-Raptor (clone 24C12); anti-pRaptor (IgG, Ser792); anti-Rictor (IgG, clone D16H9); anti-pRictor (IgG, Thr1135, clone D30A3) (all obtained from Cell Signaling), PKBα/AKT (IgG1, clone 55), and anti-pAkt (IgG1, pS472/pS473, clone 104A282) (both from BD Biosciences. To investigate integrin proteins and integrin-related signaling, detection was completed with anti-α2 (clone 2; Merck Millipore, Darmstadt, Germany); anti-β1 (clone 18; BD Biosciences); anti-integrin-linked kinase (ILK, clone 3); anti-focal adhesion kinase (FAK, clone 77, and anti-pFAK (pY397; clone 18) antibodies (all obtained from BD Biosciences). HRP-conjugated goat anti-mouse IgG and HRP-conjugated goat anti-rabbit IgG (both obtained from Cell Signaling) served as secondary antibodies. Protein visualization was completed with the Fusion FX7 system [10]. To evaluate whether IGF1 mediates mTOR activity, HepG2 cells were stimulated with 100 ng/mL IGF alone or simultaneously treated in combination with 5 and 10 nM Rapamycin. Protein was extracted after 24 h, and both mTOR and pmTOR were analyzed by Western blotting. To quantify the intensity of the protein bands, the protein intensity/β-actin intensity ratio was calculated by the GIMP 2.8 software. The original WB can be found in Appendix A. 

### 2.7. IGF1R Detection and IGF1 Stimulation

IGF1 receptor (IGF1R) expression was evaluated fluorometrically by the PE-conjugated anti-Human CD221 (IGF1R; clone1H7; BD Pharmingen) monoclonal antibody. Mouse IgG1 (clone P3.6.2.8.1; Thermo Fisher, Dreieich, Germany) was used as the control isotype. In ongoing experiments, the tumor cells were activated with LONG^®^R3IGF-1 (100 ng/mL) and adhesion, chemotaxis, and the integrin α2 (clone 12F1-H6), αV (clone 13C2), and β1 (clone MAR4) surface expression was analyzed. Adhesion to immobilized collagen and chemotaxis was also evaluated when tumor cells were serum starved for 12 h and stimulated with IGF-1 and simultaneously blocked with anti-integrin α2 (clone P1E6) or anti-integrin β1 (clone 6S6), both mouse mAb (all obtained from Merck Millipore, Burlington, MA, USA). BTT3033 (Bio-Techne, MN, USA), a selective inhibitor of the integrins α2β1, or Cyclo(RGDyK) (Selleckchem, TX, USA), targeted against integrin αVβ3, were also employed to determine whether cross-communication between IGF-evoked Akt signaling and integrin expression exists. HepG2 cells were activated with IGF and integrin α2 and β1 expression were measured by flow cytometry and compared to the integrin expression in HepG2 cells treated with the AKT inhibitor MK-2206 (Selleckchem, TX, USA).

### 2.8. Adhesion and Invasion

Tumor cell adhesion to immobilized collagen (collagen G, 400 µg/mL; Biochrom, Berlin, Germany) was also evaluated [10]. Tumor cells were exposed to either culture medium alone or to culture medium enriched with LONG^®^R3IGF-1 (100 ng/mL). The number of attached cells was counted microscopically and the mean cellular adhesion rate calculated, according to [11]. Serum-induced invasion was investigated using a Boyden double chamber system with 8 µm pore filters (Greiner Bio-One, Frickenhausen, Germany). The chamber was pre-coated with Matrigel (BD Biosciences), and 0.5 × 10^6^ tumor cells/mL were then added to the upper chamber, containing serum-free medium. The lower chamber was filled with culture medium enriched with 10% FBS. The tumor cells were either activated with LONG^®^R3IGF-1 (100 ng/mL), or they received culture medium without IGF (controls). Cells which crawled underneath the filter membrane were counted microscopically, and the mean invasion rate was calculated [11].

### 2.9. Integrin Receptor Blockade and siRNA Knock Down

Tumor cells were treated with function-blocking anti-integrin α2 (10 μg/mL; clone P1E6) or anti-integrin β1 (10 μg/mL; clone 6S6) mouse mAb (all from Merck Millipore). BTT3033 or Cyclo(RGDyK) were used to block integrin α2β1 or integrin αVβ3, respectively. Subsequently, tumor cells were subjected to the MTT-assay, the clonogenic growth assay, the cell cycle test, and the adhesion and chemotaxis assay. Integrin knockdown was also completed, using small interfering RNA (siRNA) against integrin α2 (gene ID: 3673, target sequence: 5′-CCCGAGCACATCATTTATATA-3′; Qiagen, Hilden, Germany) or integrin β1 (gene ID: 891, target sequence: 5′-AATGTAGTCATGGTAAATCAA-3′; Qiagen, Hilden, Germany). The procedure complied with the manufacturer’s protocol. Following transfection, protein expression and tumor cell growth were then explored, as described above.

### 2.10. Immunoprecipitation

Immunoprecipitation was performed using Dynabeads™ Protein G Immunopre-cipitation kit (Thermo Fisher). Briefly, cells were lysed on ice with Pierce™ IP lysis buffer containing protease inhibitor and homogenized using Precellys^®^ Evolution homogenizer. To pellet cell debris, protein lysates were centrifuged at 13,000× *g* for 20 min at 4 °C. The antibody magnetic bead complex was prepared by adding 5 µg mouse anti-human integrin β1 (Clone:18/CD29) to the magnetic beads for 10 min at room temperature. Beads were washed three times with binding and washing buffer. Antigen was immunoprecipitated by adding the protein extracts to the magnetic antibody complex for 1 h at 4 °C in a rotator mixer. The beads (Magnetic ab-ag complex) were then washed three times with washing buffer, followed by elution and denaturation of the target antigen at 70 °C for 10 min. The eluted target protein was subjected to Western blot and detected using rabbit anti-pAKT (Ser473, Clone: D9E) or mouse anti-AKT (Clone: 55/PKBa/Akt).

### 2.11. Statistics

All experiments were repeated three to five times, and the mean +/− SD was calculated. Statistical significance was evaluated with the Student’s *T*-Test. *p* < 0.05 was considered significant.

## 3. Results

### 3.1. Collagen and Matrigel Induced Cell Growth Alterations

Since many effects of the integrins are mediated by contact with the extracellular matrix, the tumor cells were incubated with soluble matrix proteins and cell growth was evaluated. Collagen and Matrigel altered the HepG2, Hep3B, and Huh7 cell growth dose-dependently. An increased number of HepG2 and Huh7 (but not Hep3B) cells was detected in the presence of 1 µg/mL collagen (*p*-values *** 0.0009, **** <0.0001, respectively), whereas exposure to 100 µg/mL collagen resulted in a significant growth reduction in HepG2, Hep3B, and Huh7 cells (Figure 1) (*p*-values *** 0.0002, **** <0.0001, **** <0.0001, respectively). Elevated tumor cell growth was observed when the cells were exposed to 10 µg/mL soluble Matrigel, with the strongest effects being exerted on the HepG2 cells (*p*-value **** <0.0001). The 1 µg/mL of Matrigel had no effect, and 100 µg/mL of Matrigel diminished the cell number of the Hep3B (but not the HepG2 and Huh7) cells (each compared to unstimulated controls) (*p*-value * 0.0423).

### 3.2. Basal Integrin α and β Expression Levels

The basal integrin α and β expression levels in the HepG2, Hep3B, and Huh7 cells show that the integrins α2, αV, and β1 were strongly expressed in all three cell lines (Figure 2). The integrins α1 and α6 were also detectable in the tumor cell membrane surface. The α5 was seen in the Hep3B and Huh7, but not in the HepG2, cells. The integrin members α3, α4, β3, and β4 were not present in any of the tumor cell lines.

### 3.3. Influence of Matrix Proteins on Integrin Subtype Expression

Significant differences in the α2, αV, and β1 expression became evident, when the tumor cells were treated with collagen. Based on flow cytometry, expression of α2, αV, and β1 on HepG2 was increased when the cells were exposed to 1 µg/mL soluble collagen or 10 µg/mL soluble Matrigel, but decreased when incubated with 100 µg/mL soluble collagen (Figure 3A). The same effect was seen with Huh7 cells, except for αV, which did not increase with added 1 µg/mL collagen or 10 µg Matrigel, and β1 which was not altered by Matrigel. In Hep3B cells, α2 was elevated with 1 µg/mL collagen or 10 µg/mL Matrigel. However, 100 µg/mL soluble collagen led to a loss of α2, αV, and β1. Therefore, to maintain the integrin stimulation in further investigation, a collagen concentration of 1 µg/mL or a Matrigel concentration of 10 µg/mL was employed. In this context, the gene expression of integrins α2, αV, and β1 were evaluated in HepG2 and Huh7 following 1 µg/mL collagen or 10 µg/mL Matrigel exposure. Significant elevation of the αV gene and suppression of the β1 gene were found in HepG2, whereas only αV was altered in Huh7 cells (Figure 3B).

Alterations in HepG2 cells related to tumor cell growth were investigated by integrin α2, αV or β1 knockdown, and by blocking integrin α2, αV, and/or β1 surface receptors (Figure 4A,B). Knockdown of α2, αV or β1 significantly reduced the HepG2 cell growth after 72 h incubation (Figure 4A) (*p*-values *** 0.0003, *** 0.0002, **** 0.0001, respectively) The knockdown of β1 resulted in a growth reduction approximately twice that for α2 knockdown (50 and 27%, respectively) (*p*-value **** <0.0001). The strongest effects were seen following αV knockdown (*p*-value **** <0.0001). Blocking the effects of α2, αV, and β1 by monoclonal antibodies or α2β1 by BTT3033 and αVβ3 by Cyclo caused a significant reduction in cell growth (Figure 4B) (*p*-values * 0.0452, **0.0034, ** 0.0014, * 0.0266, *** 0.0002, respectively). The enhancement of tumor cell growth by collagen (1 µg/mL) or Matrigel (10 µg/mL) was also suppressed when α2, αV or β1 were blocked by BTT3033 or Cyclo (Figure 4C) (*p*-values ** 0.0078, * 0.0183, respectively). Whether the tumor cells were pre-treated with Matrigel or not, BTT3033 or Cyclo induced a cell growth arrest at G2/M (Figure 4D). This suppression was verified by the clonogenic growth assay, where the Matrigel-induced increase in HepG2 clones was significantly inhibited by blocking the integrins α2, αV, and β1 with specific function-blocking antibodies, BTT3033, and Cyclo (Figure 4E) (*p*-values **** <0.0001, **** <0.0001, respectively).

### 3.4. Integrin α2, αV, and β1 Expression in the Presence of IGF1

To investigate whether IGF1R activation by the growth factor IGF1 may be relevant for the integrin-driven invasion processes, tumor cells were incubated with IGF, and adhesion and migration, along with integrin expression, were analyzed. IGF1R was expressed on all cell lines, with HepG2 exhibiting greatest expression (Figure 5). The HepG2 cell line was, therefore, used for further studies.

Stimulation with IGF1 induced a significant upregulation of the integrins α2, αV, and β1 (Figure 6A) and induced a significant increase of HepG2 cell attachment to immobilized collagen and forced invasive movement (Figure 6B–D, respectively) (*p*-values ** 0.0035, **** <0.0001, **** <0.0001, **** <0.0001, respectively). Blocking the integrins α2 and β1 with respective monoclonal antibodies or with BTT3033 or blocking αVβ3 by Cyclo counteracted the IGF1 triggered adhesion and invasion (Figure 6B–D). Knockdown of α2, αV, or β1 was also associated with a significant reduction in the HepG2 invasion (Figure 6E) (*p*-values **** <0.0001, **** <0.0001, **** <0.0001, respectively)

### 3.5. Cell Signaling Alterations Caused by IGF1 Exposure

IGF1 activation of HepG2 resulted in a distinct modification of the CDK-Cyclin axis, in AKT/mTOR signaling, as well as in integrin-related proteins (Figure 7A,B, Appendix A). Cyclin A and B, CDK2 and the phosphorylated form of CDK1 and 2 were upregulated. The mTOR, Raptor (both: total and phosphorylated) and pRictor were elevated as well. Notably, a massive increase in pAKT was seen. The pFAK, from the integrin-related signaling proteins, was significantly enhanced following IGF exposure (Figure 7A,B). The interconnection with integrin signaling was then investigated by knocking down integrins α2 and β1 in the HepG2 cell line (Figure 7A,C,D). Knocking down α2 was associated with a reduction in Cyclin A and B, pCDK1 and pCDK2, AKT, pmTOR, pRictor, and Raptor. FAK was elevated. However, pFAK and ILK were strongly diminished (Figure 7A,C). With the β1 knockdown, Cyclin A, pCDK1, Akt and, strongly, pAKT, pmTOR, and pRictor, were downregulated. Distinct loss in pFAK and ILK was also apparent (Figure 7A,D).

In a subsequent step, interaction of IGF1 signaling with mTOR was evaluated. Stimulating HepG2 with IGF1 resulted in a distinct upregulation of pmTOR which was abolished when the HepG2 cell line was treated before with the mTOR inhibitor Rapamycin (Figure 8A) (*p*-value **** <0.0001). The relevance of pAKT signaling for integrin expression was also explored. Enhanced integrin α2 and β1 expression, induced by IGF1 (*p*-values ** 0.0040, ** 0.0028, respectively), was prevented by pre-treating the tumor cells with the specific AKT inhibitor MK-2206 (Figure 8B) (*p*-values ** 0.0029, * 0.0379,** 0.0035, * 0.0360, * 0.0414, ** 0.0012, respectively).

Interaction between integrins and Akt signaling is finally verified in Figure 9. Immunoprecipitation documented that both AKT (moderately) and pAKT (strongly) cross-communicate with the integrin subtype β1.

## 4. Discussion

Cross-communication between AKT/mTOR signaling and the integrin subtypes α2, αV, and β1 in a cell model of HCC were apparent. The growth promoting AKT/mTOR pathway was connected to adhesion and invasion processes, and alterations of integrin adhesion receptors were closely related to cell cycle and growth regulation.

The integrin-AKT/mTOR-interaction has not been fully understood and is discussed controversially. Using the same cell lines as we did, inhibition of the mTOR downstream target p70-S6K1 exhibited a decrease of pFAK, followed by an altered integrin β1 expression and suppression of epithelial–mesenchymal transition [12]. Based on experiments with Huh7 cells, activating the FAK/AKT signaling pathway through integrin β4 was suggested to promote hepatocellular carcinoma metastasis [13], whereas the association between integrin β1-FAK-AKT and proliferation, migration, and invasion of hepatoma BEL-7402 cells was assumed by others [14]. There is also evidence that integrin β3 [8] or α7 [15] may be linked to AKT. Activating the HCC cell lines HepG2 and Huh7 with the integrin ligand collagen at a concentration of 1 µg/mL caused significant upregulation of the tumor cell number in our model, paralleled by an increase in the α2 and β1 integrins. These integrins might, therefore, be particularly involved in HCC cell growth control. Indeed, blockade or knockdown of α2 and β1 was associated with suppressed HepG2 cell number and generation of tumor clones, verifying this supposition. Therefore, the interaction of α2 and β1 with collagen could activate intracellular signaling cascades that promote tumor cell growth.

Differences of the mechanistic integrin regulation should be taken care of. Matrigel caused a distinct elevation of integrin α2, αV, and β1 surface expression on HepG2 cells, accompanied by an upregulation of the αV, but not of the α2 and β1 gene. Obviously, αV might be regulated transcriptionally, whereas elevation of α2 and β1 might be controlled translationally. In contrast, collagen exposure increased the αV gene (but not α2 and β1), whereas the αV receptor level on HepG2 cells remained unchanged. Presumably, the αV gene transcription caused by collagen did not further progress to the step of protein synthesis.

Several publications have already documented that integrin β1, once connected to extracellular matrix proteins, serves as a major driver of HCC metastasis [16]. The present investigation provides evidence for the first time that the integrin α2/β1–collagen interaction also activates pathways relevant for mitotic HCC progression. After binding to collagen, α2/β1 integrin was shown to activate the pro-oncogenic Yes-associated protein (YAP) in HCC cells, which correlated well with tumor progression and outcome for HCC patients [17]. HepG2 cells cultured in a combination of type IV collagen and laminin, which is in accordance with Matrigel composition, showed phosphorylation of MEK1/2 through β1 integrins [18]. MEK1/2 activation is considered a critical step during hepatocarcinogenesis and tumor proliferation [19].

Stimulating several HCC cell lines with soluble laminin (1 µg/mL) has previously been shown to enhance proliferation and it was hypothesized that particular integrin subtypes may be responsible for laminin-induced growth regulation [20]. The present investigation shows that the engagement of α2 and/or β1 integrin to the extracellular matrix may result in the formation of intracellular signaling networks facilitating cell proliferation.

Notably, diminished cell growth in the presence of highly concentrated collagen (100 µg/mL) was not only paralleled by diminished α2 and β1 expression, as expected, but also by a reduction in αV. We did not further deal with this high collagen concentration and, therefore, can only speculate on the underlying mechanism. Studies on human hepatocytes have shown that adding highly enriched collagen to cell cultures results in the formation of gel-like structures, enabling the acquisition and stabilization of an epithelial phenotype that is accompanied by a loss of mitotic activity [21]. Presumably, a similar scenario has occurred in the present culture system under 100 µg/mL collagen, with prevention of de-differentiation along with suppression of the cell cycling machinery. This may also explain why integrin αV, which does not serve as a collagen ligand (and, therefore, was not elevated in the presence of 1 µg/mL collagen), became diminished in a MET promoting milieu. Indeed, facilitating the process of MET with abrogation of tumor proliferation was shown to be accompanied by integrin αV downregulation [22].

The addition of 10 µg/mL Matrigel (1, 10, and 100 µg/mL Matrigel were tested) induced the strongest promotional effects on HepG2 cell growth. This was paralleled by an increase in both integrin α2 and β1. Slighter effects were evoked in the Huh7 and Hep3B cell lines, without alteration of integrin β1 expression. The concerted action of integrin α2 and β1 therefore seems required to induce maximum growth. This was verified by the integrin blocking studies where simultaneous blockade of α2 and β1 diminished the tumor cell number to a stronger extent than did the separate blockade of the two integrins. In sound agreement, incubating hepatic stellate cells with Matrigel enhanced the expression of the growth promoting transcription factor Ets-1 via the integrin complex α2β1, but not via α1β1 [23]. Whether integrin β1 dominates α2, as was apparent during knockdown, requires further investigation. Similar to the effects evoked by collagen, αV and β1 in HepG2 and αV exclusively in Huh7 cells were transcriptionally regulated, whereas this was not the case with integrin α2.

From a mechanistic viewpoint, α2, αV, and β1 were shown to be involved in cell cycle regulation, whereby the blockade of α2/β1 or αV arrested HepG2 cells in G2/M. How these integrin subtypes specifically act on cell cycling is not yet clear. Liu et al. assumed the switching of α2/β1 binding to collagen isoforms to establish a sequentially signaling transduction to accelerate HCC mitosis [24]. Downregulation of integrin β1 in HepG2 cells concomitantly increased the number of G0/G1 and G2/M phase cells in another study with CSN5 (the fifth subunit of COP9 signalosome) as the relevant trigger factor [25]. The integrin member αV was suggested to be a crucial target of the microRNA (miR)-122, finally leading to activation of the extracellular matrix–receptor interaction pathway and cell cycle-associated processes [26].

The protein analysis of HepG2 cells demonstrated that IGF not only activates the AKT/mTOR growth pathway, but also FAK phosphorylation. Zheng et al. observed a higher proliferation rate of HepG2 cells when cultured with liver matrix proteins. This was associated with elevated integrin β1 expression and activation of downstream pFAK [27]. Likewise, the knockdown of α2 and β1 in the present investigation provides evidence that these integrins are closely linked to pFAK. Since stimulation with IGF resulted in elevated α2 and β1 expression, in turn activating FAK, it may be assumed that IGF triggers progressive HCC growth by means of the α2/β1-pFAK-pathway. In this context, knocking down α2 or β1 led to a significant de-activation of AKT-mTOR signaling and reduced CDK phosphorylation, showing that α2/β1 is linked to molecules acting as the main modulators of cell cycling. In particular, β1 was strongly involved in de-phosphorylating AKT and mTOR, which might explain why β1 knockdown was superior to α2 knockdown in reducing HepG2 cell growth. Knockdown of integrin β1 was shown to partially attenuate the phosphorylation levels of AKT and mTOR in a stemness model of HCC [28]. Whether integrins α2 and β1 deliver signals to AKT-mTOR, or whether they receive signals from AKT/mTOR, is still not clear. Suppression of AKT by MK-2206 abrogated the elevation of α2 and β1, evoked by IGF. There is no doubt that AKT is a prominent regulator of mitotic processes, making it likely that IGF primarily influences HCC growth via AKT/mTOR with integrins α2 and β1 serving as downstream mediators. In line with this hypothesis, a β1 integrin inhibitory antibody was shown to effectively suppress the activation of both FAK and AKT in HepG2 spheroids, accompanied by proliferation inhibition and apoptosis induction [29]. However, Guo et al. concluded that β1 regulates HCC development via the AKT pathway [30], leaving the question of primary activation by AKT or integrins open.

When analyzing the adhesive and invasive behavior of HepG2, α2, αV, or β1 appear to serve as the primary target structures activating FAK and initializing motile crawling, since increased adhesion and invasion by IGF could be counteracted by blocking α2, αV, or β1. This means that IGF-triggered alteration of HCC motility could depend on the level of integrin expression. Further signals might then be transmitted from the integrin–FAK axis to IGF1R, driving the invasion cascade forward by additional AKT-mTOR signaling. Further investigation is required to exactly determine the order of the signaling events in the course of metastatic HCC progression, but the principal interconnection between integrin subtypes and AKT was established. The integrin subtype β4 was shown to promote HCC migration and invasion by altering AKT phosphorylation [9]. Interaction of integrin β4 with the epidermal growth factor receptor (EGFR) on the HCC cell surface was suggested to activate AKT, leading to the development of lung metastases in vivo [31]. Finally, attachment of HCC cells during the process of invasion was shown to elevate integrin β1 and subsequently transmit signals to AKT [32]. A similar mechanism seems to have occurred in the present investigation with the integrins α2, αV, and β1 driving HCC adhesion and invasion, and AKT/mTOR serving as down-stream targets. In support of this, α2/β1-dependent adhesion of HCC cell lines to collagen was shown to activate IGF1R downstream [17].

The AKT-mTOR pathway was documented to be responsible for building the CDK-cyclin complex and activating the CDK-dependent signaling [33]. In the present investigation, pCDK1, pCDK2 along with cyclin A and B were all downregulated following α2 knockdown, whereas β1 knockdown was somewhat associated with reduced AKT-mTOR phosphorylation. Whether the integrin β1 subtype closely interacts with AKT-mTOR and the integrin α2 subtype predominantly cooperates with members of the CDK-cyclin family is not yet clear. It is, however, speculated that α2 at least partially bypasses communication with AKT-mTOR. In support of this it was shown that the expression of CDK1 is required for an integrin-dependent stimulation of prostate cancer cell migration [34]. CDK1-integrin crosstalk was also reported to initiate FAK phosphorylation in Schwann cells [35] and to regulate cell adhesion during cell cycling [36].

## 5. Conclusions

HCC growth, proliferation, and invasion are controlled by a fine-tuned network between α2/β1-FAK signaling, the AKT-mTOR pathway, and the CDK–Cyclin axis. Presumably, the α2 subtype communicates with CDK-Cyclin, whereas β1 contacts AKT-mTOR as its primary target. Concerted blockade of the integrin α2/β1 complex may, therefore, open an option to prevent progressive dissemination of this tumor type.

## Figures and Tables

**Figure 1 cancers-14-02430-f001:**
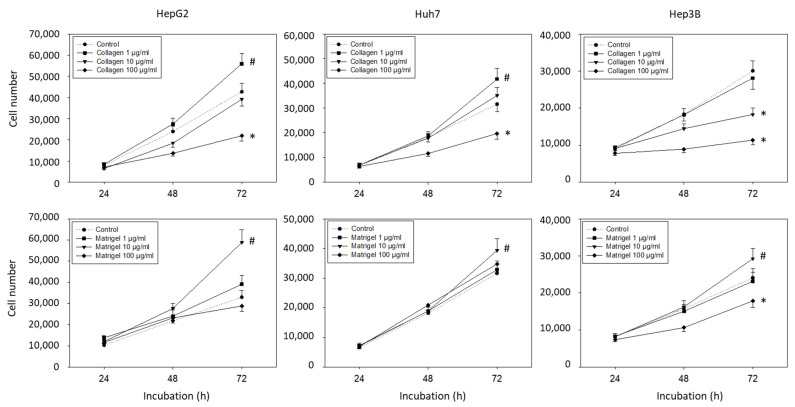
Cell growth evaluation (mean +/− SD). Hepatocellular carcinoma cell number in response to 0 (control), 1, 10, and 100 µg/mL collagen or Matrigel. MTT analysis was carried out after 24, 48, and 72 h. * significant downregulation, # significant upregulation, each compared to untreated controls (*n* = 5).

**Figure 2 cancers-14-02430-f002:**
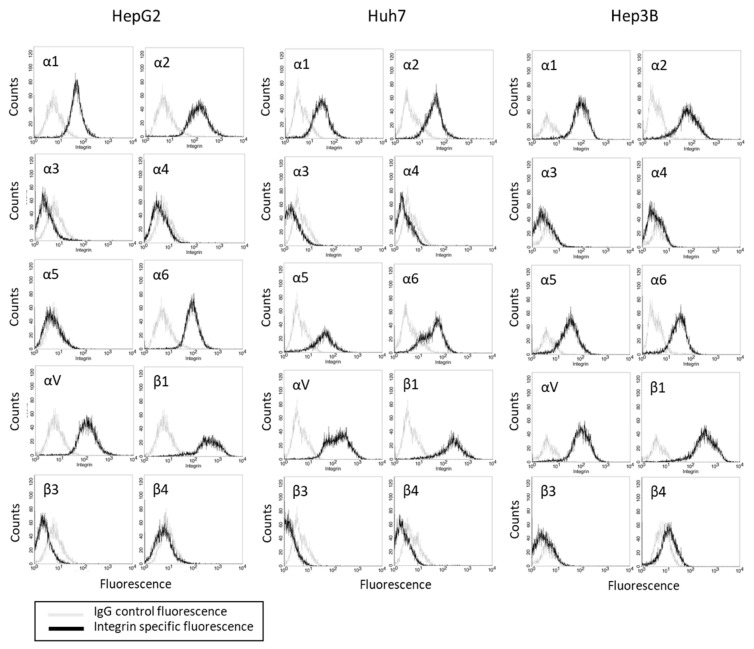
Integrin α and β expression profile on HepG2, Huh7, and Hep3B cells. Cell number (Counts) versus fluorescence intensity is depicted (one representative of three separate experiments). Black line = specific fluorescence, grey line = isotype control.

**Figure 3 cancers-14-02430-f003:**
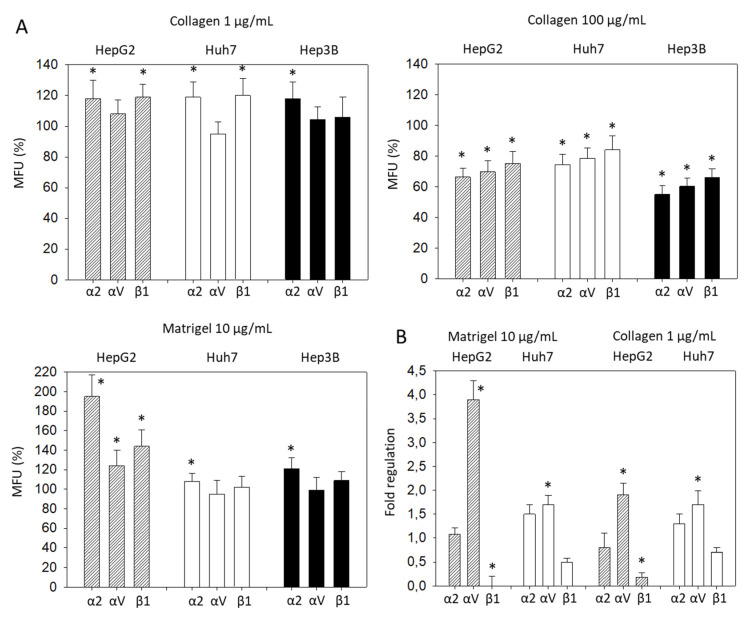
(**A**) Integrin α2, αV, and β1 expression level evaluated 72h after exposing HCC cells to collagen (1 µg/mL, 100 µg/mL) or Matrigel (10 µg/mL). All fluorescence values (MFU) are related to untreated controls set to 100%. Means ± SD of *n* = 4. * significant difference to the controls set to 100%; (**B**) Gene expression of integrin α2, αV, and β1, depicted as fold regulation relative to the control (control = 1). * indicates significant difference.

**Figure 4 cancers-14-02430-f004:**
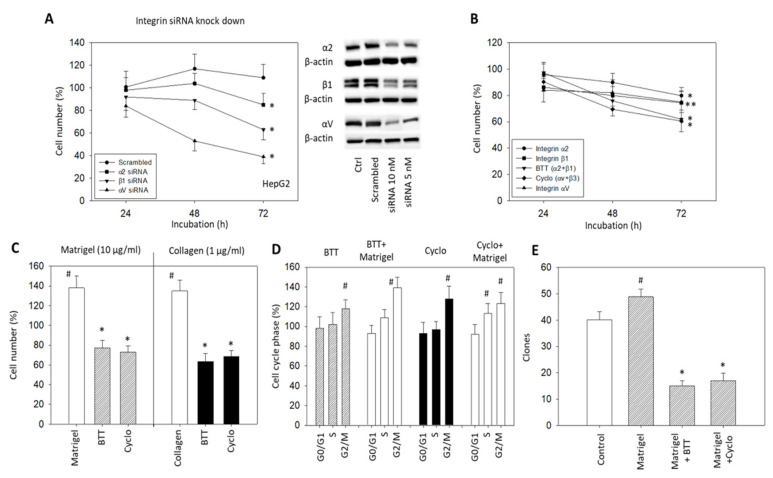
(**A**) HepG2 cell growth in response to α2, β1 or αV knockdown with small interfering RNA (siRNA) (Western blot at right); (**B**) HepG2 cell number in response to integrin blocking antibodies targeted against integrin α2, β1, αV, α2β1 (BTT), or αVβ3 (Cyclo). The respective MTT assay was completed after 24, 48, and 72 h. Cell number of non-blocked controls was set to 100%. * (**A**) indicates significant difference as marked. * (**B**) indicates significant difference to untreated controls (*n* = 3); (**C**) HepG2 cell growth in the presence of Matrigel (10 µg/mL) or collagen (1 µg/mL) and subsequent to Matrigel or collagen plus the blockade of integrin α2β1 with BTT or αV with Cyclo. Cell number was evaluated after 72 h by the MTT assay and related to untreated controls set to 100%. (*n* = 3); (**D**) Cell cycle analysis of HepG2 cells under integrin α2β1 (BTT) or αV (Cyclo) blockade, or under stimulation with Matrigel (10 µg/mL) and blockade with integrin α2β1 (BTT + Matrigel) or αV (Cyclo + Matrigel). Percentage of cells in the G0/G1-, S-, and G2/M-phase is indicated (all related to respective controls set to 100%) (*n* = 3); (**E**) Clonogenic growth in the presence of Matrigel (10 µg/mL), in the presence of Matrigel plus integrin α2β1 blockade (Matrigel + BTT) or in the presence of Matrigel plus integrin αV blockade (Matrigel + Cyclo). Control cells remained untreated. # significant up-regulation, * significant down-regulation to untreated controls (*n* = 4).

**Figure 5 cancers-14-02430-f005:**
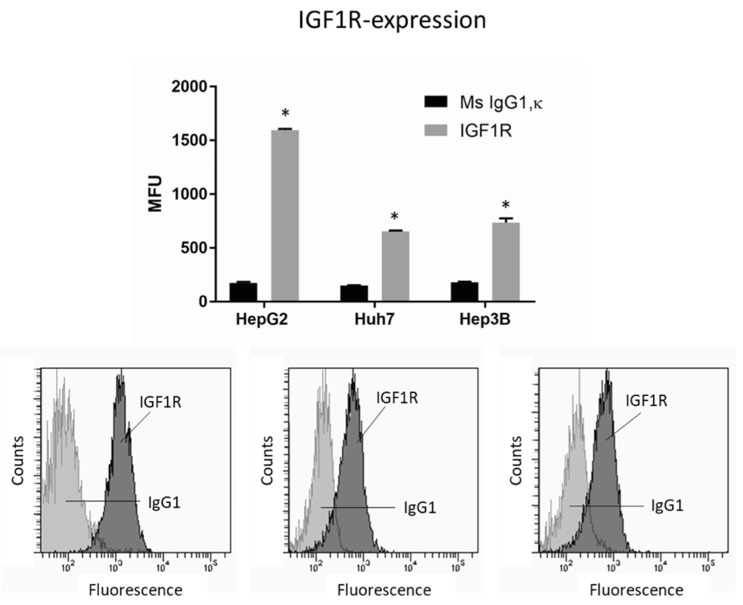
IGF1R expression level in HepG2, Huh7, and Hep3B cell lines. MFU depicts mean fluorescence units. * indicates significant difference to Ms IgG1 (*n* = 3). Flow cytometry curves are representative for one of the three experiments.

**Figure 6 cancers-14-02430-f006:**
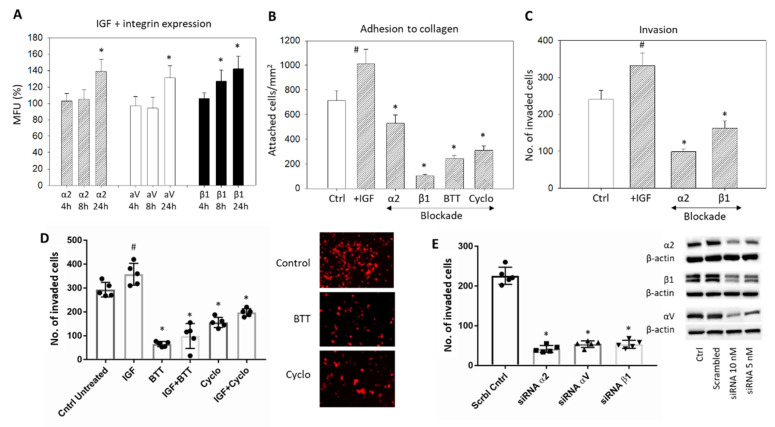
(**A**) IGF1 stimulated expression of the integrins α2, β1, and αV in HepG2 cells, evaluated after 4, 8, and 24 h. The figure depicts fluorescence intensity, related to untreated controls set to 100%. * significant difference (*n* = 3); (**B**) HepG2 cell adhesion to immobilized collagen after stimulation with IGF1 plus blockade with specific antibodies targeting α2 or β1, α2β1 (BTT) or αV (Cyclo). Controls remained untreated. Mean adhesion +/− SD is shown. # significant upregulation, * significant downregulation, compared to the untreated controls (*n* = 4); (**C**) Invasion of HepG2 cells after stimulation with IGF1 plus blockade with specific antibodies targeting α2 or β1. Cell number is related to untreated controls; (**D**) Invasion of HepG2 cells (mean +/− SD) after stimulation with IGF plus blockade by BTT (blockade against α2β1) and Cyclo (blockade against αVβ3). Controls remained untreated. # significant difference to the untreated control, * significant downregulation to the control (*n* = 3). Right panel represents stained invaded cells (x200 objective; control versus BTT or Cyclo treatment); (**E**) HepG2 cell invasion in response to α2, β1, or αV knockdown with small interfering RNA (siRNA) (Western blot at right).

**Figure 7 cancers-14-02430-f007:**
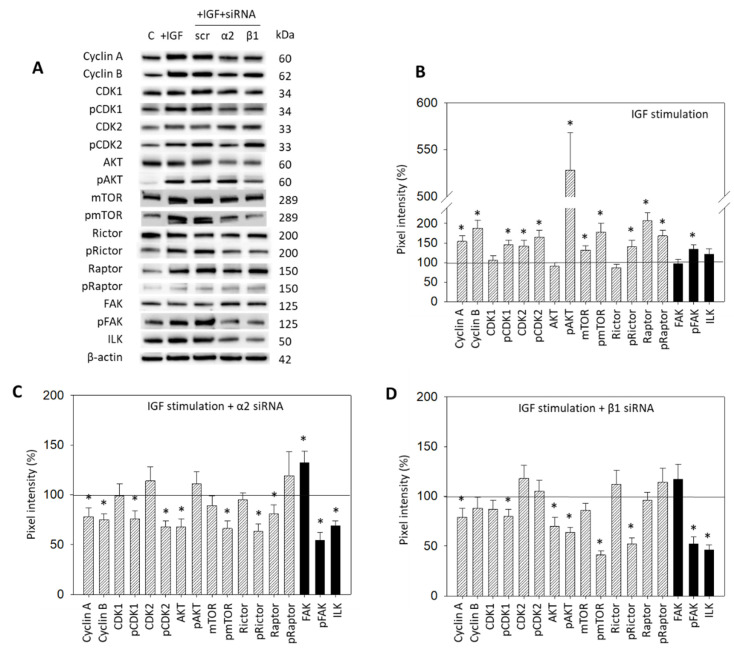
(**A**) Protein profile of cell cycle regulating proteins (AKT, mTOR, Rictor, Raptor, all total and phosphorylated, “p”), CDK1 and 2 (total and phosphorylated, “p”), Cyclin A and B, FAK, pFAK and ILK. HepG2 cells were either stimulated with IGF or stimulated with IGF and treated with an integrin α2 or β1 specific siRNA (scr = scrambled siRNA). Following siRNA transfection protein data of integrin α2 and β1 are shown on the right side of Figure 4A. Controls (**C**) received cell culture medium alone. One representative experiment of three is shown; (**B**) Quantification of the protein intensity in IGF1 treated versus non-treated HepG2 cells by the pixel density analysis, all related to the unstimulated 100% controls. * significant difference to controls; (**C**,**D**): Pixel density analysis of the protein level in HepG2 cells following α2 (**C**) or β1 (**D**) siRNA knock-down. Values are given in percentage, related to the 100% control. * significant difference to controls.

**Figure 8 cancers-14-02430-f008:**
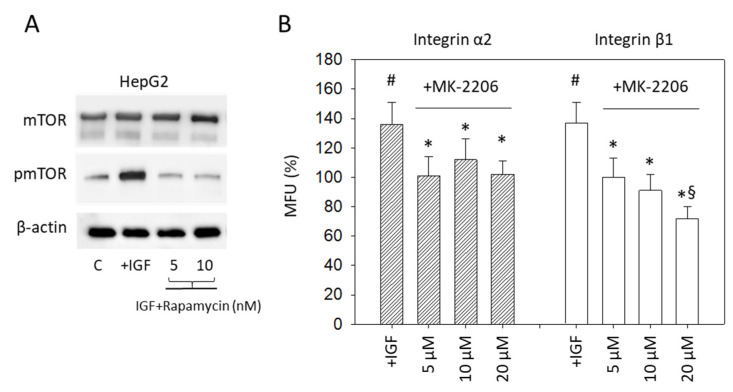
(**A**) mTOR and pmTOR protein expression. HepG2 cells were stimulated with IGF1 (IGF), remained untreated (C), or mTOR was blocked by the mTOR-inhibitor Rapamycin and then treated with IGF1. Protein analysis was accompanied by a β-actin loading control. One representative experiment of three is shown; (**B)** Surface expression of the integrin subtypes α2 and β1 on HepG2 cells. Tumor cells were either exposed to IGF1 (+IGF) or stimulated with IGF1 under AKT blockade with MK-2206. Detection was completed by FACS analysis. Mean fluorescence values (MFU) are related to untreated controls set to 100%. # indicates significant upregulation, compared to cells not exposed to IGF. * indicates significant downregulation to IGF stimulated cells. § indicates significant downregulation to cells not exposed to IGF1 (*n* = 3).

**Figure 9 cancers-14-02430-f009:**
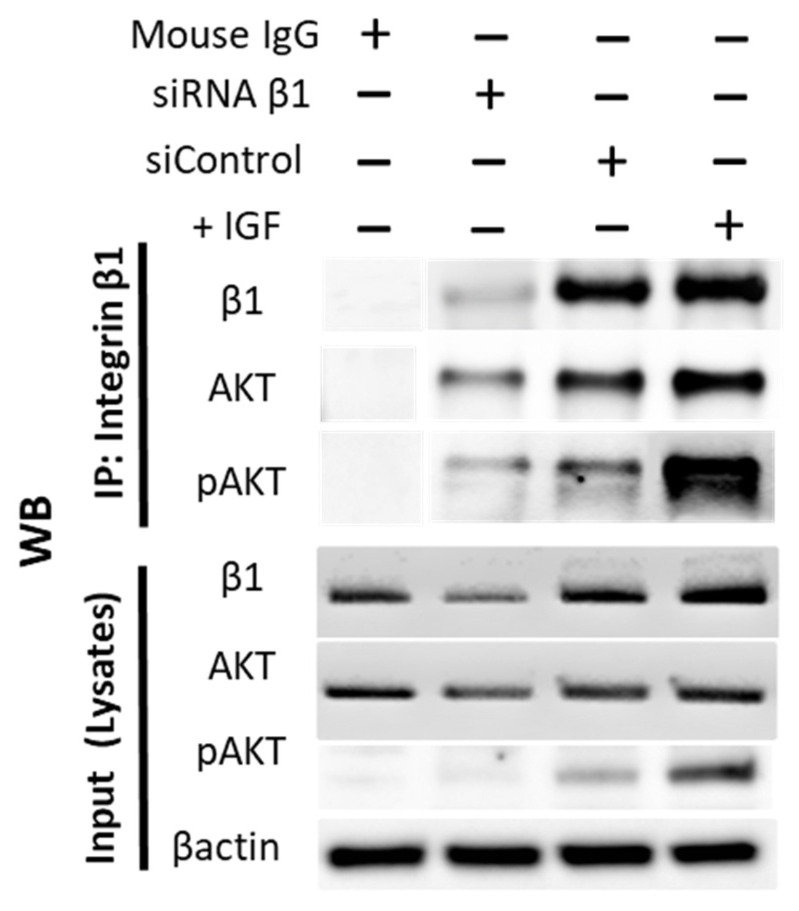
Integrin β1 interacts with AKT. IP was performed wherein integrin β1 was pulled down using mouse anti Human CD29 (Clone:18/CD29). Following SDS-PAGE electrophoresis and WB, the separated proteins were probed with antibodies against β1, AKT, and pAKT. Lysate from cells treated with IGF1 (+IGF) demonstrated strong interaction of pAKT with integrin β1. Lysate from cells treated with siRNA β1 showed knockdown efficiency of β1 protein. Mouse IgG was used as a negative con-trol. A total of 20 µg of cell lysate was used as input. IP, immunoprecipitation; WB, Western blotting.

## Data Availability

The data presented in this study are available on request from the corresponding author.

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
