# Peer review of "Integrin α2 and β1 Cross-Communication with mTOR/AKT and the CDK-Cyclin Axis in Hepatocellular Carcinoma Cells"

_cancers, 2022, doi:10.3390/cancers14102430_

Round 1
Reviewer 1 Report
The manuscript entitled " Integrin α2 and β1 cross-communication with mTOR/AKT and the CDK-Cyclin axis in hepatocellular carcinoma cells" (cancers-1711480 had provided some revising again the previous suggestions although the major points were not adequat enough. No further questions could be adressedfurther.
Reviewer 2 Report
The article "Integrin α2 and β1 cross-communication with mTOR/AKT and the CDK-Cyclin axis in hepatocellular carcinoma cells" (Juratli et al.) is really interesting, well written, data are adequately presented and the conclusions are clearly supported by the results. The minor corrections I suggested and other changes suggested by the referees had been taken into account, and the changes are shown at the new version. I recommend to accept the new version of article for its publication.
Reviewer 3 Report
The authors addressed the different comments and the paper can be published in the present form.
This manuscript is a resubmission of an earlier submission. The following is a list of the peer review reports and author responses from that submission.
Round 1
Reviewer 1 Report
In current manuscript (cancers-1431466, the authors claimed Integrin α2 and β1 cross-communication with mTOR/Akt and the CDK-Cyclin axis in hepatocellular carcinoma cells. This is an connecting story with poor evidence to link each other. First, Integrin activated mTOR/AKT pathway is well known. The authors just provided cell number and and protein expression level to demenstrate the well-known phenamina. No other approach and no further mechanism prediction to show that. For example, the author did not show how Integrin α2 and β1 "cross-communication" to activate mTOR/AKT? How activated integrin complex activated mTOR/AKT (Genetic or non-genetic)?Nest, mTOR/AKT signals activated Cyclins is also well-known. Again, the auther claimed the connection through simple cell number and protein expression level. No other althernative approach and mechanism prediction to show that. I think the manuscript is not completed for publishing.
Reviewer 2 Report
The article from Juratli MA et al.: “Integrin α2 and β1 cross-communication with mTOR/Akt and 2 the CDK-Cyclin axis in hepatocellular carcinoma cells” is well written and the methods and the results are very interesting and easy to read . However, there are some questions that need to be explained, and some changes at the discussion and conclusions seem to be needed.
Invasive potential and metastatic potential are not equivalent terms. The metastatic capacity implies extracellular matrix degradation, along with higher migration rates. HepG2 and Huh7 are non tumorogenic cell lines with no metastatic potential in nude mice. Hep3B is tumorogenic and capable to metastatize. It is a cell line that carries HBV, so biosafety level 2 is necessary to manage this cell line (it is sometimes recommended also for Huh7. Biosafety levels are missing at material and methods). If the authors aimed to demonstrate changes related with HCC progressive disease by using a cell model to evaluate changes in cell invasivity, it is not clear to me why used HepG2 to migration assays, instead of using Hep3B and invasion assays. Using HepG2 cells, the authors demonstrate changes in cell migration, but this does not means invasive nor metastatic potential.
At the introduction (starting at line 63) the following can be read: “The process of vascular invasion and HCC metastasis is firmly controlled by adhesion receptors of the integrin family, in particular integrins of the β1 subtype e.g. α1β1 and α5β1 [3]. Recently, the integrins β3 [4] and β4 [5] have been demonstrated to not only promote HCC migration but invasion as well”. Since it is a clear difference at the introduction, it is confusing the use of the terms “migration”, “invasion” and “metastasis” along the paper, since equivalence between “migration” and “invasion” can be frequently inferred. Cells able to migrate are not necessary able to invade and, therefore, be able to metastatize. The authors demonstrate changes in cell migration but no changes in invasion, since this would require a different experimental approach: invasion implies degradation of the extracellular matrix. The classical Boyden chamber assay is used for migration assays, not for invasivity assays. The use of matrigel in the Boyden chamber, for instance, is necessary for this kind of analysis. So, authors cannot conclude changes on invasion. At the conclusions, it can be read “HCC growth, proliferation, and invasion are controlled by a fine tuned network between α2/β1-FAK signaling, the AKT-mTOR pathway and the CDK-Cyclin axis. Presumably, the α2 subtype communicates with CDK-Cyclin, whereas β1 contacts AKT-mTOR as its primary target. Concerted blockade of the integrin α2/β1 complex may, therefore, open an option to prevent progressive dissemination of this tumor type”. It is confusing, since the authors did not demonstrate changes in cell invasion, just migration. Of course, an increase in the capacity of migration is necessary to start the invasive and metastatic program, but cells capable of migrate might not be capable of invade and disseminate. HepG2 cells are neither tumorogenic nor have metastatic potential, although this can be modified, for instance, by means of different culture conditions or induced genetic changes. Changes at the text must be made to accurately adjust to these concepts and changes at the conclusions are necessary.
Reviewer 3 Report
Juratli MA et al. showed that stimulation of HCC cell lines by collagen or Matrigel increased proliferation and migration and that this increase is dependent on the expression of integrins alpha2, alphaV et beta1. Treatment of the HCC cell lines with IGF1 led to an increase expression of the integrins studied and this expression seems dependent on AKT signalling pathway.
Comments
1/ Introduction
- Detail more what’s already known regarding the link between the PI3K/AKT/mTOR pathway and HCC.
- Develop what is known about CDK/Cyclin dysregulation and HCC.
2/ results
- Figure 2: show the 2 other experiments in supp data.
- For all figures: show exact p values
- Figure 3a and b : discuss the differences between the gene expressions levels of the different integrins analysed and their cell surface expression. For example, the increase of alpha V cell surface expression when cells are treated with collagen or Matrigel is not statistically significant whereas alpha V gene expression is the highest among the 3 integrins in treated cells.
- Title 3.3, 3.4, and 3.5 : change the titles to a more precise title
3/ Discussion:
- Discuss more what is already known regarding the link between these signalling pathways and proliferation, migration and invasion.
- line 414 to 427 : How could HepG2 and Huh7 have gone through a mesenchymal to epithelial transition after treatment with high concentrations of collagen since these two cell lines are epithelial-like ?